# Agility Skills, Speed, Balance and CMJ Performance in Soccer: A Comparison of Players with and without a Hearing Impairment

**DOI:** 10.3390/healthcare11020247

**Published:** 2023-01-13

**Authors:** Hakan Yapici, Yusuf Soylu, Mehmet Gulu, Mehmet Kutlu, Sinan Ayan, Nuray Bayar Muluk, Monira I. Aldhahi, Sameer Badri AL-Mhanna

**Affiliations:** 1Department of Coaching Education, Faculty of Sport Sciences, Kirikkale University, Kirikkale 71450, Turkey; 2Department of Physical Education and Sports, Faculty of Sport Sciences, Tokat Gaziosmanpasa University, Tokat 60250, Turkey; 3Department of Coaching Education, Faculty of Sport Sciences, Hitit University, Corum 19000, Turkey; 4ENT Department, Faculty of Medicine, Kirikkale University, Kirikkale 71450, Turkey; 5Department of Rehabilitation Sciences, College of Health and Rehabilitation Sciences, Princess Nourah bint Abdulrahman University, P.O. Box 84428, Riyadh 11671, Saudi Arabia; 6Department of Physiology, School of Medical Sciences, Universiti Sains Malaysia, Kubang Kerian 16150, Kelantan, Malaysia

**Keywords:** soccer, hearing-impaired, athletic performance, skill, physical fitness

## Abstract

This study investigates the differences in agility, speed, jump and balance performance and shooting skills between elite hearing-impaired national team soccer players (HISP) and without-hearing-impairment elite soccer players (woHISP). Players were divided into two groups, the HISP group (*n* = 13; 23.5 ± 3.1 years) and the woHISP group (*n* = 16; 20.6 ± 1.4 years), and were tested in three sessions, seven apart, for metrics including anthropometrics, speed (10 m, 20 m and 30 m), countermovement jump (CMJ), agility (Illinois, 505, zigzag), T test (agility and shooting skills), and balance. The results showed that 30 m, 20 m and 10 m sprint scores, agility/ skills (sec), shooting skills (goals), zigzag, Illinois, and 505 agility skills, and countermovement jump scores were significantly lower among players with hearing impairments (*p* < 0.05). There were no significant T test differences between HISP and woHISP (*p* > 0.05). The HISP showed right posterolateral and posteromedial, and left posterolateral and posteromedial scores that were lower than the woHISP group (*p* < 0.05). Anterior scores were not significantly different between each leg (*p* > 0.05). In conclusion, the HISP group showed higher performance scores for speed (10 m, 20 m and 30 m), CMJ, agility (Illinois, 505, zigzag) and *T* test (sec and goals), but not balance. Hearing-impaired soccer players are determined by their skill, training, and strategy, not their hearing ability.

## 1. Introduction

World Health Organization data reveal that there are about 466 million people with hearing impairments worldwide [1]. As stated by the International Committee of Sports for the Deaf, deafness is the capacity to hear sound only at a frequency of 55 dB or higher in the better ear [2]. Hearing-impaired soccer players, just like all soccer players, are expected to have good speed, agility and balance. These skills enable soccer players to move quickly and efficiently on the field, change direction quickly, and maintain their balance when receiving or passing the ball or making a shot at a goal [3,4].

Soccer game characteristics involve intermittent high-intensity physical and technical tactical demands such as rapid changes of direction, jumps and dribbling [5,6]. Because of increasing game requirements, players must improve their physical skills based on different soccer match demands [7,8]. As a result, during the period of the season, players can usually maintain or increase their overall aerobic and anaerobic fitness [9]. Nevertheless, monitoring football’s internal and external loads, mainly acute workload, differs between game periods [10]. Soccer-specific abilities refer to the specific physical, technical and tactical skills required to perform well. Training programs have been designed to improve motor abilities such as agility, sprinting and balance to enhance overall physical performance in soccer [11,12]. However, game performance has been affected by various obstacles, including physical, mental or psychological challenges [5,13].

Physical obstacles such as hearing loss, rather than a physical disability such as poor performance or injury, may affect athletes’ performance compared to those without a hearing impairment [14]. Long-term hearing loss may also significantly alter sensory processing and lead to motor deficiencies [15]. From this point, various physical and physiological factors also affect how well hearing-impaired soccer players perform during training and games. However, hearing-impaired soccer players have been less able to perform due to poor body composition, and less aerobic endurance, strength, balance and agility compared to without-hearing-impairment players [16,17]. Therefore, developing anthropometrics and physical and physiological characteristics of hearing-impaired footballers is also necessary to increase performance to respond to the developments in soccer.

Even though previous studies have shown that hearing-impaired soccer players have differences in some physical performance parameters, there is no evidence that these characteristics are directly related to hearing ability, except for balance related to a damaged vestibular system [18]. Limited studies on hearing-impaired soccer players have focused on acute physical performance response [16] or compared hearing-impaired and non-hearing-impaired soccer players in laboratory fields [18], and female soccer players’ biomechanical characteristics [19]. Therefore, more studies are needed comparing elite soccer groups without hearing impairments and similar hearing-impaired groups’ results for performance outside the laboratory. When the performance responses of hearing-impaired soccer players were analyzed, we realized that more studies in this field are needed. We hypothesized that elite hearing-impaired national team soccer players (HISP) might show higher performance based on technical abilities and some physical parameters, except balance ability, than without-hearing-impairment soccer players (woHISP).

## 2. Materials and Methods

### 2.1. Participants

This study was conducted with a cross-sectional design. An initial power analysis (G*Power, University Duesseldorf, Duesseldorf, Germany) was performed to determine the sample size [20]. A total of 29 players were separated into two groups, both of which comprised elite soccer players: HISP (*n* = 13, age: 23.5 ± 3.1, height: 177.8 ± 5.1, weight: 72.8 ± 3.7, body mass index: 23.1 ± 1.9) and woHISP (*n* = 16, age: 20.6 ± 1.4, height: 178.4 ± 6.1, weight: 72.5 ± 8.7, body mass index: 23.2 ± 2.9). The inclusion criteria were as follows: (i) all physical evaluations conducted before the start of the season; (ii) no injuries at the time of the evaluations; and (iii) no injuries in the month prior to the evaluations (Figure 1). The HISP included the ICSD criteria to compete, a minimum hearing loss of 55 dB in the better ear or both ears, and being active athletes of the Turkish Deaf Sports Federation, competing at the national level at the time of the study. However, the HISP group had European, World and Olympic champions and medal-winners.

Before the testing, the volunteers were asked to provide information about their training experience, education level and hearing loss level. The sign language interpreter informed the HISP groups. This study was conducted at Kirikkale University, Sports Sciences Faculty, according to the principles outlined in the Declaration of Helsinki. Athletes were fully informed about the procedures of this study, and all volunteers signed a written consent form before study participation. This study was approved by the Kirikkale University Non-Invasive Researches Ethics Committee (Date: 10 December 2020, number: 7 December 2020).

### 2.2. Measurements

#### 2.2.1. Anthropometric Measurements

Body height (BH), body mass (BM) and composition measurements were performed using a bioelectrical impedance analysis (BIA) and body composition analyzer (Tanita Body Composition Analyzer BC 418 professional model, Japan). The subjects’ heights were measured with an anthropometrics rod set on the day before performance tests, to the nearest 1 cm. We preferred using BIA testing in estimating body fat percentages (BFP), based on some research studies showing validity. Related literature stated that BIA is a suitable alternative for estimating body fat percentages when subjects are within a “normal” body fat range. However, there is a tendency for BIA to overestimate body fat in lean subjects and underestimate body fat in obese individuals.

#### 2.2.2. Y Balance Test

The Y balance test was performed to evaluate postural control. Participants were allowed a maximum of six trials., with three successful attempts for each access direction. After three successful attempts in each direction, the rater recorded the maximum and average distance. In addition, the reach distance was recorded to the nearest 0.5 cm [21].

#### 2.2.3. 10, 20 and 30 m Sprint Test

A Brower Timing System (USA) was placed 10, 20 and 30 m away from the starting point with a photocell device (0.01-s precision). When the athlete felt ready for the sprint test, they were asked to start with their preferred foot in front. The times between the start and finish gates at 10 m (first electronic timing gate) and 20 m (second electronic timing gate) were recorded. After a 10 min rest interval, the measurement was repeated three times, and the best performance was recorded for further analysis [22]. 

#### 2.2.4. Illinois Test

The participant began the test by lying face down on the floor behind the starting line with arms at their sides and head to the side or forward. With the “start” command, the participant stood up and ran the test track, which consisted of three cones lined up in a straight line at intervals of 5 m in width, 10 m in length and 3.3 m in the middle section. Two trials were run, a rest period of 3 min was given between trials, and the best score was recorded. If the participant did not run the course by the instructions, did not reach the finish lines, did not complete the course or moved any cone, they were disqualified [23].

#### 2.2.5. 505 Agility Test

After a 10 m approach run at the start of a 15 m track, the participants were asked to run 5 m forward and backwards as quickly as possible. During testing, the time between two passes was recorded. The best result from repeated measurements was recorded in seconds [24]. 

#### 2.2.6. Zigzag Test

The test for agility, acceleration, deceleration and balance consisted of 4 × 5 m sections set at 100° angles. The test started with the exit of the participants from the photocell door (point A). They ran to point B, returned from point B, reached the middle point C again and, finally, after passing through the photocell at point D, the test was terminated. Each participant repeated the test three times, and the best score was recorded [22].

#### 2.2.7. Countermovement Jump (cm)

Participants were asked to assume a full squat position with their hands on their waist and were told to jump as high as possible with maximum force, without making any springing movements from the knees. The participants were asked to have the same positions during jumping and landing on the platform again. During the test, athletes were told not to move forward, backward or sideways, and keep their hands on their waist. This test was conducted on each athlete three times, and the best score was used for analysis [24,25].

#### 2.2.8. Agility and Shooting Skill Test

This test was applied to measure and determine the agility and mobility status of the athletes as well as their ability to score a goal. A Brower Timing System photocell device with the same start and end point was placed at the test site. After the necessary information was given to the subjects, the time began when they passed through the starting gate. The athletes kicked ball number 1 in the middle with their right foot, then ran to ball number 2, where they kicked the ball with their left foot. The subjects ran from here to ball number 3 on the right, kicked this with their right foot, ran to number 4, kicked the ball with their left foot, then ran to the starting point with their back turned, and time stopped when they passed through the finish gate. Participants were asked to score goals by hitting the four balls into the goal. The subjects hit four balls. The athletes’ times were recorded in seconds. The raw time score was re-evaluated as the total skill and agility score. If the subjects managed four goals, three goals, two goals or one goal, 1.00s, 0.75 s, 0.50 s or 0.25 s, respectively, were subtracted from the completion time [26].

### 2.3. Procedures

The testing sessions were performed each day for seven days at the university’s Exercise Physiology Laboratory and gymnasium. The first session of tests included measurements of body composition before breakfast. Sprint tests were determined after a 15 min standardized soccer warm-up. Agility tests were carried out during the second test session. Lastly, balance tests consisted of three different movement directions in the third test session: anterior, posteromedial and posterolateral. Participants were instructed to adhere to the following guidelines before all tests: dress in sports apparel, avoid vigorous exercise and alcohol consumption 24 h before testing and be ready to test correctly hydrated. All participants were instructed to perform each test with their maximum effort via verbal encouragement throughout each trial. All participants were tested in a specific order to standardize the testing process: weight, height, body composition, balance, sprint, and agility. Standardized procedures were followed for each assessment test and are published in the Procedures section below.

### 2.4. Statistical Analysis

Data were represented as mean ± standard deviation (SD). The data were evaluated to see whether they had normal distribution or not. The Shapiro–Wilk test was not verified for the normality assumption [27]. Comparison of physical and sports characteristics, speed, agility and balance results were analyzed using the Mann–Whitney U test. To analyze the relationship between independent variables, Spearman’s correlation rho efficient test was used. The following scale was used to assess the magnitude of the Spearman’s correlations and their differences: 0.00 to 0.10, trivial; 0.11 to 0.29, small; 0.30 to 0.49, moderate; and 0.5 to, large [28]. Cohen’s (d) standardized effect size was used to determine the magnitude of differences, using the following thresholds [26]: 0.0 to 0.2, trivial; 0.2 to 0.6, small; 0.6 to 1.2, moderate; 1.2 to 2.0, large; >2.0, very large. A value of *p* < 0.05 was considered statistically significant. Variables were analyzed using IBM SPSS for Windows Version 21.0. (IBM Corp., Armonk, NY, USA). The American Psychological Association (APA) 6.0 style was used to report statistical differences [29]. 

## 3. Results

In terms of educational level, a total of three (23%) hearing-impaired subjects (HISP group) had bachelor’s degrees and 10 (77%) had high and secondary school degrees. Measurement results in HISP and woHISP groups, and physical and training characteristics are shown in Table 1.

### 3.1. Physical and Sports Experience Characteristics

There were no significant differences between body height (*p* = 0.774; d = 0.11, trivial ES) and body mass (*p* = 0.554; d = trivial ES) of the HISP and woHISP groups. The sports experience of HISP was significantly greater than woHISP (*p* = 0.000; d = 2.48, very large ES). The BFP (%) of the HISP group (9.9 ± 1.2%) was significantly lower than the woHISP group (*p* = 0.000; d = 1.89, large ES) (Table 1).

### 3.2. Sprints and Agility/Skill Variables

In the HISP group, sprint scores were 4.91 ± 0.21 s, 2.45 ± 0.10 s and 1.75 ± 0.10 s, respectively. In the woHISP group, sprint scores were 5.70 ± 0.52 s, 2.84 ± 0.25 s and 1.91 ± 0.20 s, respectively. The 30 m (*p* = 0.000; d = 2.06, very large ES), 20 m (*p* = 0.000; d = 2.05, very large ES) and 10 m (*p* = 0.017; d = 1.01, moderate ES) sprint scores of the HISP group were significantly lower than those of the woHISP group (Table 1).

The *T* test scores were not different between the HISP and the woHISP groups (*p* = 0.895; d = 0.42, small ES). However, the agility/skill (sec) of the HISP group was significantly lower than the woHISP group (*p* = 0.002; d = 1.54, large ES). Moreover, the shooting skill (goals) scores of the HISP group were significantly lower than the woHISP group (*p* = 0.000; d = 2.01, very large ES). The zigzag (*p* = 0.000; d = 2.58, very large ES), Illinois (*p* = 0.000; d = 3.80, very large ES) and 505 agility skill (*p* = 0.002; d = 1.60, large ES) scores of the HISP group were significantly lower than those of the woHISP group. The jumping scores of the HISP group were significantly higher (*p* = 0.002; d = 1.20, large ES) than the woHISP group (Table 1).

### 3.3. Y Balance Test Results

There were no significant differences between anterior Y balance scores of the HISP and woHISP groups for the right (*p* = 0.272; d = 0.531, small ES) and left (*p* = 0.392; d = 0.30, small ES). However, the right posterolateral (*p* = 0.000; d = 4.30, very large ES) and right posteromedial (*p* = 0.000; d = 2.30, very large ES), and left posterolateral (*p* = 0.000; d = 636, very large ES) and left posteromedial (*p* = 0.000; d = 3.49, very large ES) Y balance scores of the HISP group were significantly lower than those of the woHISP group (Table 1).

### 3.4. Correlation Test Results in the HISP Group Are Shown in Table 2

There were positive correlations between the 30 m and 20 m sprint scores (*p* < 0.01; r = 0.995, large), zigzag and 30 m (*p* < 0.01; r = 0.730, large), 20 m (*p* < 0.01; r = 0.690, large), agility skill and 10 m (*p* < 0.05; r = 0.584, large) and Illinois and 10 m (*p* < 0.05; r = 0.592, large). As 505 agility skills scores increased, jumping scores decreased (*p* < 0.05; r = −0.633, large). As shown in the results in Table 2, there were positive correlations between body fat and right posteromedial Y balance scores (*p* < 0.05; r = 0.567, large), left posterolateral (*p* < 0.05; r = 0.567, large), Illinois and left posterolateral Y balance (*p* < 0.05; r = 0.594, large) and right posterolateral and zigzag (*p* < 0.05; r = 0.631, large).

## 4. Discussion

This study aimed to determine sprint, agility, balance, jumping and skill-based performance response differences between HISP and woHISP groups. The main findings of this study revealed that the HISP group’s sports experience, BFP, 10 m, 20 m, 30 m, agility/skill, agility and shooting skill tests, zigzag Illinois, 505, jumping, right and left posterolateral and posteromedial scores were significantly better than woHISP. 

The current study results indicated that the HISP group’s BFP (%) was largely higher than the woHISP group. The deaf Czech national soccer team have been shown to have higher BF% but lower free fat mass than without-hearing-impairment players [17]. Comparing hearing female players and hearing-impaired soccer players, hearing players have significantly different waist- and calf-circumferences, and waist–hip ratios [19]. Some studies have found changes in body composition throughout the season, demonstrating how these changes may be significantly influenced by factors such as training intensity, exposure to match time, or diet [30,31]. Players who are lean and have a high percentage of muscle mass may have an advantage in positions that require speed and agility. On the other hand, players who are heavier and have a higher percentage of body fat may have an advantage in positions that require strength and power. It is important to note that soccer players should aim to have a body composition that allows them to perform at their best and meet the demands of their position or role on the field. This may involve maintaining a healthy weight, sufficient muscle mass and balanced body fat distribution. Players should also be mindful of their overall health and wellness and aim to maintain a healthy diet and exercise routine that helps them meet their physical goals.

Regarding physical fitness results, our study showed that while the HISP group had better 30 m, 20 m and 10 m sprints, agility, agility skills and jumping performance than woHISP, balance performance was reduced. During soccer games, high-intensity movements can be classified as actions requiring quick acceleration (10 m sprint), actions demanding maximum speed (30 m sprint), or actions requiring agility [22]. Additionally, it is well known that professional athletes perform better during sprints (10, 20, 30 and 40 m) in games than their less-skilled counterparts [32].However, these sprints typically last between two and four seconds and cover distances of less than 20 m [33]. Sprint ability is essential for soccer players, as it allows them to accelerate quickly and reach high speeds when running with the ball or chasing down opponents. This study has important findings, and the HISP group performance revealed significantly better agility abilities and skills with the ball to woHISP. While the classic definition of agility is the simple capacity for rapid direction changes, agility is a crucial performance variable comprising perceptual and decision-making characteristics such as visual scanning and anticipation in soccer [34]. Considering the effect of the reaction time on athletic performance, hearing-impaired athletes scored better in visual reaction time [35]. Visual reaction time is an essential factor in athletic performance, as it can affect an athlete’s ability to react to changes in the game or competition. The visual reaction time or technical skills of deaf athletes will depend on their abilities and training, and are not necessarily related to their hearing status. Our main finding demonstrated that CMJ performance was higher in the HISP group than the woHISP group. Neuls et al. [17] reported that Czech national team soccer players had lower scores than the first league, but these were not significant. In contrast to our results, Soslu et al. [14] performed a study on repeated countermovement jump tests for volleyball and basketball players who were hearing-impaired and deaf. The results showed that deaf players’ scores, in particular, were reduced for jumps and jump heights, the force produced, the acceleration at the time of the jump and the jump velocity compared to non-deaf players. There need to be more studies in the hearing-impaired soccer player context to compare different soccer players. 

In this study, balance tests scores recorded during the Y balance test showed a significant decrease in the HISP group compared to the woHISP group. Additionally, a significant difference was found between the posterolateral and posteromedial in both legs, with a very large effect. These findings align with a previous study on national deaf basketball players, who demonstrated lower balance performance than hearing-impaired players [36]. As a result of the vestibular, visual and proprioceptive systems being injured, the balance may be affected negatively. Without a visual component, the balance will likely be impaired in deaf individuals who already have damage to this region [37]. Balance is crucial in soccer, as it allows players to maintain their footing and control the ball while moving on the field. Factors affecting balance performance in soccer include strength, coordination and proprioception.

The second aim of this study was to analyze the correlation of agility with speed, jumping and balance. Our results revealed large correlations between agility and speed. Our results are similar to previous studies reporting strong correlation between speed, agility and agility/skills [22,38,39]. Compared to non-elite soccer players, elite players have higher levels of physical fitness and talent [40]. The relationship between speed and agility in soccer is complex, as both abilities are important for success on the field. However, agility is often more important in soccer, as it is more relevant to the movements and actions players need to perform during a game. Both speed and agility can be improved through training, and many soccer players incorporate specific drills and exercises into their training programs to improve these abilities. 

Another finding of this study was the negative correlation between jumping and 505 agility performance. Our results are similar to the previous study of Köklü et al. [38], which reported a significant negative relationship (r = −0.769, *p* = 0.01) between jumping and agility (angle 100°) without the ball. Strong correlations between COD and CMJ would be expected, given that CMJ is an example of a long stretch-shortening cycle action (500 ms) that requires time to create a force for propulsion [41,42]. In the acceleration phase, where much force must be produced quickly to overcome the inertia in both vertical and horizontal orientation, lower limb power (represented by CMJ and cross hop) can, therefore, be considered a determinant of change in direction [43]. Our study revealed strong correlations between balance and agility. A previous study supports our results; dynamic balancing performance was recently shown to be significantly correlated with COD performance [44], and another study revealed that moderate correlations indicated that dynamic balance is an influential factor in agility, but is not the main limiting factor [41]. It is essential to recognize that hearing impairment is just one factor that can impact an athlete’s performance, and it is not fair to generalize the abilities of hearing-impaired athletes. Each athlete is unique and has their strengths and challenges. The most important thing is to provide support and resources to help hearing-impaired athletes reach their full potential, regardless of their performance relative to non-deaf athletes. Considering the sample group of this study, the fact that they are Olympic-level athletes causes them to show better results than the results in other studies.

The current study has some limitations. A comparison has been made between the hearing-impaired group of athletes and others who were not similar in their sports experience and age. The soccer players without a hearing impairment had lower ages and sports experience. For this reason, further research is necessary to better match the larger sample size of elite adult soccer players in the same class, age category and sports experiences for the with- and without-hearing-impairment groups.

## 5. Conclusions

In conclusion, the current study highlighted that the HISP group had higher speed (10 m, 20 m, and 30 m), agility (zigzag, Illinois, 505), CMJ and agility skills (sec and goals). Balance performance was lower in the HISP group than the woHISP group, including the posterolateral and posteromedial of the right and left legs. These results show that hearing-impaired athletes have the potential to perform at the same level and even better than non-hearing-impaired athletes with appropriate training methods and planning. Additionally, hearing-impaired athletes are a special population with very different anthropometric and physical characteristics. Therefore, coaches must provide adequate anthropometrics and physical performance profiles for players with different characteristics and training stimuli during seasons. It seems to be worth investigating possible factors influencing players’ workload.

## Figures and Tables

**Figure 1 healthcare-11-00247-f001:**
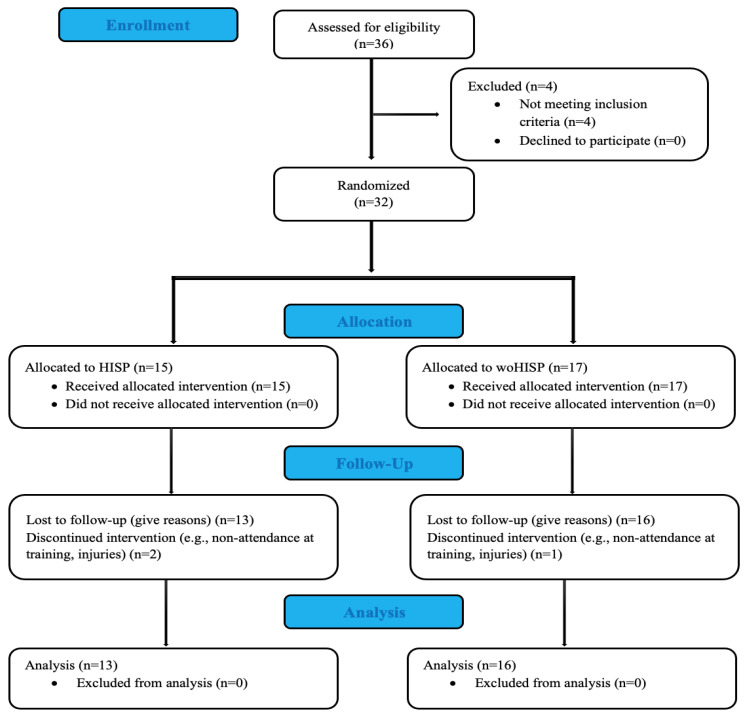
CONSORT flow diagram.

**Table 1 healthcare-11-00247-t001:** Measurement results in soccer HISP groups and woHISP groups.

		Group 1 (HISP) (*n* = 13)		Group 2 (woHISP) (*n* = 16)		CI 95%				
		Mean	SD	Mean	SD	Lower	Upper	Cohen’s *d*	Descriptor	*p* *
** *Physical and sports experience characteristics* **										
Body height (cm)		177.84	5.17	178.43	6.19	−5.0	3.8	0.11	Trivial	0.774
Body mass (kg)		72.86	3.72	72.54	8.77	−5.0	5.7	0.05	Trivial	0.554
Sports experience (year)		12.30	3.30	6.25	1.00	−2.9	1.3	2.48	Very Large	0.000 *
BFP (%)		9.76	1.17	11.86	1.05	−1.1	−0.5	1.89	Large	0.000 *
** *Sprints and agility/skill variables* **										
30 m (Sec.)		4.91	0.21	5.70	0.52	−0.5	−0.3	2.06	Very Large	0.000 *
20 m (Sec.)		2.45	0.10	2.84	0.25	−0.3	−0.1	2.05	Very Large	0.000 *
10 m (Sec.)		1.75	0.10	1.91	0.20	−0.3	−0.1	1.01	Moderate	0.017 *
T Test		10.33	0.14	10.45	0.38	−1.1	−0.3	0.42	Small	0.895
Agility/Skill (Sec.)		11.45	0.30	12.18	0.60	−1.1	−0.5	1.54	Large	0.002 *
Agility and Shooting Skill Tests (goals)		10.59	0.22	11.38	0.51	−1.1	−0.5	2.01	Very Large	0.000 *
Zigzag (Sec.)		6.10	0.23	7.60	0.79	−1.9	−1.0	2.58	Very Large	0.000 *
Illinois (Sec.)		16.79	0.43	18.71	0.57	−2.3	−1.5	3.80	Very Large	0.000 *
505 Agility (Sec.)		2.40	0.17	2.86	0.37	−0.7	−0.3	1.60	Large	0.002 *
CMJ (cm)		51.10	1.77	48.10	3.07	1.0	4.9	1.20	Large	0.002 *
** *Y-balance test* **										
	Anterior	73.69	12.13	76.68	6.07	−10.1	4.1	0.31	Small	0.272
Right foot	Posterolateral	66.00	13.08	110.00	6.16	−51.6	−36.4	4.30	Very Large	0.000 *
	Posteromedial	72.07	14.33	99.87	9.38	−36.9	−18.7	2.30	Very Large	0.000 *
	Anterior	71.38	8.65	74.06	9.49	−9.7	4.3	0.30	Small	0.392
Left foot	Posterolateral	70.61	7.83	112.25	4.95	−46.6	−36.7	6.36	Very Large	0.000 *
	Posteromedial	66.69	13.09	101.62	5.42	−42.3	−27.6	3.49	Very Large	0.000 *

* = *p* < 0.005.

**Table 2 healthcare-11-00247-t002:** Correlation test results in soccer HISP groups.

Variables (r)										Y-Balance Test–Right			Y-Balance Test–Left		
	% Fat	30 m	20 m	10 m	Zigzag	*T* Test	Illinois	505	CMJ	Right Ant **	Right PL **	Right PM **	Left Ant **	Left PL **	Left PM **
**% Fat**	1														
**30 m.**	−0.109	1													
**20 m.**	−0.095	0.995 **	1												
**10 m.**	0.350	0.264	0.248	1											
**Zigzag**	0.149	0.730 **	0.690 **	0.229	1										
**T Test**	0.062	0.466	0.482	0.584 *	0.216	1									
**Illinois**	0.203	0.387	0.370	0.592 *	0.374	0.054	1								
**505**	−0.337	0.332	0.311	−0.316	0.515	0.203	−0.262	1							
**CMJ**	0.243	−0.172	−0.164	0.171	−0.125	−0.070	0.080	−0.633 *	1						
**Right Ant ****	−0.085	0.069	0.078	−0.290	0.002	0.098	−0.394	0.070	0.226	1					
**Right PL ****	0.236	0.396	0.385	0.127	0.631 *	0.066	0.211	0.438	−0.294	0.297	1				
**Right PM ****	**0.567 ***	0.262	0.285	0.391	0.141	0.463	0.191	−0.154	0.184	0.479	0.514	1			
**Left Ant ****	0.089	0.127	0.107	−0.068	0.106	0.113	−0.037	−0.090	0.320	0.82 **	0.169	0.533	1		
**Left PL ****	**0.584 ***	−0.003	−0.042	0.330	0.083	−0.266	0.594 *	−0.480	0.120	−0.177	0.128	0.356	0.248	1	
**Left PM ****	0.401	0.222	0.218	0.234	0.487	0.230	0.003	0.126	0.206	0.545	0.78 **	0.67 *	0.447	0.060	1

** *p* < 001, * *p* < 0.005, Ant: anterior, PL: posterolateral, PM: posteromedial.

## Data Availability

The data presented in this study are available on request from the corresponding author. The data are not publicly available due to restrictions on privacy.

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
