# Peer review of "Agility Skills, Speed, Balance and CMJ Performance in Soccer: A Comparison of Players with and without a Hearing Impairment"

_healthcare, 2023, doi:10.3390/healthcare11020247_

Round 1
Reviewer 1 Report
The authors present a good research paper.
- The relevance of the topic: Good.
- Introduction: Good.
- Methodology: Can be improved.
- Results: Good.
- Discussion: Can be improved.
However, ACCEPT AFTER MINOR REVISION. In general, the paper follows an adequate structure and correct scientific support and can be published considering some limitations. The study is interesting in the field of football and disability. However, there are a series of limitations that should be considered.
In the first place, carry out a review of the existing literature related to the subject, being essential to inquire into the MPDI – Healthcare journal itself, since there are papers related to its manuscript that can help to improve it. Therefore, include those references, if any, especially from the last five years. In addition, recommend reading some papers related to the topic of football and disability:
Mawani, I. K., & Chiluba, B. C. (2020). Assessment of Non-Communicable Diseases Awareness Among Pupils with hearing impairment at Munali High School in Lusaka, Zambia. Indonesian Journal of Disability Studies, 7(1), 19-27.
Sobko, I. (2015). An innovative method of managing the training process of qualified basketball players with hearing impairment. Journal of Physical Education and Sport, 15(4), 640.
Szymczyk, D., Drużbicki, M., Dudek, J., Szczepanik, M., & Snela, S. (2012). Balance and postural stability in football players with hearing impairment. Balance, 3, 258-263.
Specific comments.
Title. The title of the manuscript is correct.
Abstract. Incorporate in the summary, a more precise sentence of the results.
Introduction. This section presents the problem in a coherent and clear manner with the correct support of the scientific literature. However, it is convenient to update the references, since there are different documents related to the subject and no mention is made, and it would even be interesting to mention the different existing studies related to football and disability. Also, it could be a future study of review. Some bibliographical references are attached to carry out the section on the practice of alternative sports as a tool for improving health:
Basakci Calik, B., Bas Aslan, U., Aslan, Åž., & Erel, S. (2019). Relationship between balance and co-ordination and football participation in adolescents with intellectual disability. South African Journal for Research in Sport, Physical Education and Recreation, 41(2), 1-9.
Fitzpatrick, D., Thompson, P., Kipps, C., & Webborn, N. (2021). Head impact forces in blind football are greater in competition than training and increased cervical strength may reduce impact magnitude. International journal of injury control and safety promotion, 28(2), 194-200.
Kitchin, P. J., & Crossin, A. (2018). Understanding which dimensions of organisational capacity support the vertical integration of disability football clubs. Managing sport and leisure, 23(1-2), 28-47.
Muñoz-Jiménez, J., Gámez-Calvo, L., Rojas-Valverde, D., León, K., & Gamonales, J. M. (2022). Analysis of Injuries and Wellness in Blind Athletes during an International Football Competition. International Journal of Environmental Research and Public Health, 19(14), 8827.
Methods. Modify the method section and incorporate the sections: Design.
- Study design. To write the design section, we recommend that you take some of the following methodologists as references.
Ato, M., López-García, J. J., & Benavente, A. (2013). A classification system for research designs in psychology. Annals of Psychology, 29(3), 1038-1059.
Montero, I., & León, O.G. (2007). A guide for naming research studies in psychology. International Journal of Clinical and Health Psychology, 7(3), 847-862.
Results. Summary of study data and table are correct.
Discussion. The section Discussion is correct.
Conclusion. Differentiate the discussion of the main conclusions of the study. To do this, you must create this section. And modify the limitations of the study and locate them in said section at the end. Also, they must be direct, and highlight the main contributions of the study.
References. They should be reviewed and updated according to the publication standards. There are many errors in the references. Therefore, correct them and adapt them to the magazine's regulations.
Author Response
We are very grateful for your patience and the opportunity of revising the paper once again.
Further to the provided recommendations, we have implemented the following changes.
In response to the comments of the Editor 1:
Comments
Dear authors, The works is relevant in a field with scarce research. However, there are some issues that need to be fixed before a further consideration. Please address all comments made by reviewers. Also, please ensure that Sample size calculation can be replicated by other which is not the case. Finally, we previously suggested using effect sizes which your reported in statistical analysis. However, those values were not reported in the results section. Please complete accordingly. Thank you
Response: Sample size calculation was added
In response to the comments of the Reviewer 1:
Incorporate in the summary, a more precise sentence of the results.
Response:
Abstract
The abstract has been written structured way again.
Introduction
The introduction has been reorganized with up-to-date references specific to the subject. The physical and physiological characteristics of hearing-impaired and without hearing-impairment athletes are presented. In the last section, different studies on football are mentioned. The abstract has been written structured way again.
Methods
The method section has been reorganised with the CONSORT diagram. Added G*Power analysis for sample size calculation.
Conclusion
The conclusion section has been rewritten.
References
References have been updated.

Reviewer 2 Report
As per notes attached

Author Response
We are very grateful for your patience and the opportunity of revising the paper once again.
Further to the provided recommendations, we have implemented the following changes.
In response to the comments of the Reviewer 1:
Response:
Abstract
The abstract has been written structured way again.
Introduction
The introduction has been reorganized with up-to-date references specific to the subject. The physical and physiological characteristics of hearing-impaired and without hearing-impairment athletes are presented. In the last section, different studies on football are mentioned. The abstract has been written structured way again.
Methods
The method section has been reorganised with the CONSORT diagram. Added G*Power analysis for sample size calculation.
The statistical method was elaborated.
The methodology was redesigned as suggested.
Primary references for all tests were used.
Results
Cohen's d was presented. And the results section has been reorganised.
Discussion
The discussion section has been rewritten.
Results are discussed in chronological order. Appropriate studies are added, and the results are described.
Conclusion
The conclusion section has been rewritten.
Practical applications are included.
References
References have been updated.

Round 2
Reviewer 2 Report
In view of the adjustments that were made to the manuscript, I consider it ready to be published, Sincerely
Author Response
dear editor, your suggested edits have been made.
